# Potent Antitrypanosomal Activities of 3-Aminosteroids against African Trypanosomes: Investigation of Cellular Effects and of Cross-Resistance with Existing Drugs

**DOI:** 10.3390/molecules24020268

**Published:** 2019-01-12

**Authors:** Charles O. Nnadi, Godwin U. Ebiloma, Jennifer A. Black, Ngozi J. Nwodo, Leandro Lemgruber, Thomas J. Schmidt, Harry P. de Koning

**Affiliations:** 1Institute of Pharmaceutical Biology and Phytochemistry (IPBP), University of Münster, Pharma Campus Corrensstraße 48, D-48149 Münster, Germany; charles.nnadi@unn.edu.ng; 2Department of Pharmaceutical and Medicinal Chemistry, Faculty of Pharmaceutical Sciences, University of Nigeria Nsukka, Enugu 410001, Nigeria; ngozi.nwodo@unn.edu.ng; 3Institute of Infection, Immunity and Inflammation, College of Medical, Veterinary and Life Sciences, University of Glasgow, Glasgow G12 8TA, UK; godwin4godwin@gmail.com (G.U.E.); Leandro.LemgruberSoares@glasgow.ac.uk (L.L.); Harry.De-Koning@glasgow.ac.uk (H.P.d.K.); 4Department of Applied Biology, Kyoto Institute of Technology, Kyoto 606-8585, Japan; 5The Wellcome Trust Centre for Molecular Parasitology, Institute of Infection, Immunity and Inflammation, University of Glasgow, Glasgow G12 8TA, UK; jennifer.stortz@glasgow.ac.uk; 6Department of Cell and Molecular Biology, Ribeirão Preto Medical School, University of São Paulo, Ribeirão Preto 14049-900, Brazil

**Keywords:** African trypanosomiasis, *Trypanosoma brucei brucei*, *Trypanosoma congolense*, mode of action, resistant strains

## Abstract

Treatment of animal African trypanosomiasis (AAT) requires urgent need for safe, potent and affordable drugs and this has necessitated this study. We investigated the trypanocidal activities and mode of action of selected 3-aminosteroids against *Trypanosoma brucei brucei*. The in vitro activity of selected compounds of this series against *T. congolense* (Savannah-type, IL3000), *T. b. brucei* (bloodstream trypomastigote, Lister strain 427 wild-type (427WT)) and various multi-drug resistant cell lines was assessed using a resazurin-based cell viability assay. Studies on mode of antitrypanosomal activity of some selected 3-aminosteroids against *Tbb* 427WT were also carried out. The tested compounds mostly showed moderate-to-low in vitro activities and low selectivity to mammalian cells. Interestingly, a certain aminosteroid, holarrhetine (10, IC_50_ = 0.045 ± 0.03 µM), was 2 times more potent against *T. congolense* than the standard veterinary drug, diminazene aceturate, and 10 times more potent than the control trypanocide, pentamidine, and displayed an excellent in vitro selectivity index of 2130 over L6 myoblasts. All multi-drug resistant strains of *T. b. brucei* tested were not significantly cross-resistant with the purified compounds. The growth pattern of *Tbb* 427WT on long and limited exposure time revealed gradual but irrecoverable growth arrest at ≥ IC_50_ concentrations of 3-aminosteroids. Trypanocidal action was not associated with membrane permeabilization of trypanosome cells but instead with mitochondrial membrane depolarization, reduced adenosine triphosphate (ATP) levels and G_2_/M cell cycle arrest which appear to be the result of mitochondrial accumulation of the aminosteroids. These findings provided insights for further development of this new and promising class of trypanocide against African trypanosomes.

## 1. Introduction

Animal African trypanosomiasis (AAT), a severe wasting disease of domestic livestock, is caused predominantly by *T. congolense*, *T. vivax* and *T. b. brucei* [1]. To date, it has remained a major threat to livestock production in sub-Saharan Africa [2,3]. With a high mortality rate of 20–50% within months of infecting cattle compounded by the lack of vaccine, and the steady increase in reported cases of drug resistance, AAT has remained a threat to food security in the affected regions [4,5]. It is endemic in 37 sub-Saharan countries where about 50 million cattle are at risk of contracting the disease, while deterring the importation of horses, high-yield dairy cattle and other non-indigenous domestic animals that are particularly vulnerable to AAT, to the further detriment of agricultural production [6]. To combat AAT, an estimated 35 million doses of trypanocides per year are used [7] in the prevention and treatment of the disease, still leaving over two thirds of the cattle population in these areas vulnerable to infection [3]. The economic consequences of AAT are evident from losses estimated at $4.5 billion per year [8], and considering the increasing failure of the few available drugs, now reported in 21 African countries [9,10], this is projected to rise further. In the past, the focus has been predominantly on the control and elimination of human African trypanosomiasis (HAT, or sleeping sickness). However, with HAT targeted for elimination by the World Health Organization (WHO) and patient numbers rapidly decreasing due to sustained efforts by international and national agencies [11,12], there is a need to refocus attention on identifying novel, potent, safe and affordable remedies for the corresponding livestock condition, given that the main control strategy for AAT is chemotherapy. The worrying epidemiological trends of AAT, in addition to the reported resistance of trypanosomes to diminazene (the most widely used trypanocide for AAT [13]), necessitates an alternative and efficient remedy, and natural products have often shown promise in this regard. Steroid alkaloids, present in the Apocynaceae, Buxaceae, Solanaceae and Liliaceae families, are an emerging class of trypanocides, with the 3-aminosteroids as the leading class of steroid alkaloids, displaying particularly strong antitrypanosomal activities [14,15] relative to other chemical classes. Clearly, there is a need to explore this for further development. Following our previous findings on lead identification and subsequent refinement of the quantitative structure-antitrypanosomal (QSAR)/cytotoxic activities relationship of steroid alkaloids from *Holarrhena africana* (Apocynaceae) against *T. brucei* species and L6 mammalian myoblast [16], there is a need to further study their activities against AAT-causing trypanosomes, including their mode of trypanocidal activity and whether cross-resistance with diminazene is likely to occur.

Determining the mode of action of a drug is imperative to understand the interaction of the biomolecule in the context of its use, since this will influence literally all further steps of the drug discovery and development process [17]. The antimicrobial actions of natural compounds are usually complex and the reported cellular or biochemical effects are often not the primary causes of phenotypic observations, due to the compounds interacting with off-target proteins and multiple targets [18,19,20], especially in trypanosomatids, which are characterized by complex and unusual biochemical processes [21]. However, such mechanistic studies are necessary to give further insights into their antitrypanosomal action, which, supplemented with the theoretical findings on 3D-QSAR [16], may be used to optimize such compounds from hits to leads and further on to new and specific drugs. Thus, the compound’s potency, its target within the parasites’ physiology and the potential to withstand the development of resistance, especially cross-resistance with the existing set of drugs, have all to be considered to identify and develop new antiparasite chemotherapy.

Here, against the backdrop of our continued effort towards the development of 3-aminosteroid lead compounds, we report the trypanocidal activities of some 3-aminosteroids against *Trypanosoma congolense* (*Tc*-IL3000, Savannah type strain), wild-type strain of *Trypanosoma brucei brucei* (*Tbb* 427WT) and some resistant strains derived from the wild type and further present observations on the mode of action of two selected 3-aminosteroids against *T. brucei* species.

## 2. Results

### 2.1. Pentacyclic 3-Aminosteroids Are More Active against Tc-IL3000 Than Tbb 427WT

The in vitro activities (IC_50_, µM) of the 3-aminosteroids **1**–**10** against *Tbb* 427WT and Tc-IL3000 and their chemical structures are shown in Table 1. All the compounds tested showed moderate activities (IC_50_, 1.6–4.8 µM) against *T. b. brucei*. In contrast, their effects against Tc-IL3000 covered a much wider range, including much more potent in vitro activities, in the mid-nanomolar range (IC_50_, 0.045–20.8 µM); most compounds (6/10) were more potent against *T. congolense*, in a few cases dramatically so. **10** was approximately twice as potent against Tc-IL3000 as the most widely used AAT trypanocide, diminazene aceturate, and was 10 times better against this species than the control HAT trypanocide, pentamidine. Generally, pentacyclic 3-aminosteroids (**6**–**10**) were more active (IC_50_ < 1.6 µM) against Tc-IL3000 than the tetracyclics (**1**–**5**) with IC_50_ > 2.9 µM, and displayed a much greater differentiation between the two trypanosome species.

### 2.2. Cross Resistance of 3-Aminosteroids with Standard Trypanocides

The aminosteroids were retested on a panel of well-documented *T. b. brucei* strains that display resistance or increased sensitivity to specific clinical and experimental trypanocides: from TbAT1-KO, the TbAT1/P2 transporter has been deleted, rendering it strongly resistant to diminazene and slightly resistant to pentamidine and arsenicals [22]; B48 was derived from TbAT1 by exposure to pentamidine and gained strong resistance to pentamidine and arsenicals [23]; ISMR1 is resistant to isometamidium and cross-resistant to diminazene [24]; R0.8 is resistant to cAMP phosphodiesterase inhibitors [25]; from AQP2/3-knockout (KO) the locus with AQP2 and AQP3 was deleted, rendering it resistant to pentamidine [26]; and AQP1-3 KO was derived from the AQP2/3 knockout by additionally deleting the TbAQP1 gene, causing hypersensitivity to inhibitors of trypanosome alternative oxidase (TAO) [27,28].

The in vitro activities against drug resistant cell lines of *T. brucei* were largely comparable to the wild-type strain; even when differences occurred, sensitivity was mostly within 2-fold relative to control wild-type cultures. Interestingly, however, compounds **3** and **5**, displayed a significant level of resistance with either one or both of the two cell lines lacking the P2/TbAT1 aminopurine transporter, which reached statistically significance (*p* < 0.05) (Table 2). It is interesting to note that these are the only two 3-amino congeners in this set with the amino function in position 3α, indicating that such aminosteroids might be substrates for this famously versatile aminopurine/diamidine transporter [29,30,31] and that the recognition is stereo-specific. However, this moderate loss of activity is less than the generally high levels of resistance to the control drugs diminazene and pentamidine in the strains lacking P2/TbAT1, indicating that the uptake of aminosteroids **3** and **5** is not as completely dependent on P2/TbAT1 than it is for the aromatic diamidines. Importantly, no cross resistance was observed with the 3β-aminosteroids—an essential requirement for new trypanosomiasis drugs—except **7**, which displayed a low but significant loss of sensitivity to all strains except the AQP-KO trypanosomes. Based on the results of the in vitro activity and the cross resistance profile of all the compounds studied, 3β-holaphyllamine (**1**) and 3β-*N*-methylholaphyllamine (**2**) were chosen for further studies on their mode of action.

### 2.3. Effects on Growth Patterns of T. b. brucei Cells after Short and Long Duration of Exposure

The growth patterns of *T. b. brucei* cells incubated with **1** and **2** in Hirumi’s modified Iscove’s medium (HMI-9) growth media were compared with untreated cells. Both **1** and **2** caused a slow-onset growth arrest of the cells at concentrations at or above their respective IC_50_ values (Figure 1) with no chances of recovery from the early effects of the compounds, even after only a brief exposure of 2 h (panels C and D). However, cells incubated with either compound at half their IC_50_ concentrations recovered from the effects and resumed growth comparable to cells in the absence of treatment.

### 2.4. Propidium Iodide (PI) Assay of Cellular Integrity

The time of onset of anti-trypanosomal action of compounds **1** or **2** on *T. b. brucei* was determined by propidium iodide (PI) assay based on the principle that damaged cells become fluorescent as PI dye begins to enter them and binds to nucleic acids. As PI is not taken up by intact trypanosomes, this can only occur when the plasma membrane is compromised as a result of loss of cellular integrity caused directly or indirectly by **1** or **2**. The effects of varying concentrations (1×, 2× and 4× the IC_50_ values) of **1** and **2** on the cell viability were monitored over a 6 h period (Figure 2). Both compounds showed slow but concentration-dependent trypanosomal membrane disruption effects. The time of onset of action was observed to be within 3–4 h of incubation, with peak effects achieved within 4 h.

### 2.5. Influence of 3-Aminosteroids on DNA Content/Cell Cycle of Wild-Type T. b. brucei

Treatment with several 3-aminosteroids resulted in alterations in cellular morphology as described above. Fluorescence-activated cell sorting (FACS) flow cytometric analysis was carried out to determine the DNA content and, hence, establish the effects of 3-aminosteroids on *T. b. brucei* cell cycle stages (G_1_, S, G_2_/M or cytokinesis) (Figure 3, and Appendix A for gating strategy and other details). The results showed a gradual and very similar decrease in the population of the G_1_ cell population and an increase in the G_2_/M phase for **1** and **2**, both at concentrations of 4× EC_50_. There was also an observable effect on S cell cycle stage by either compound as the population of DNA in the phase, although appearing relatively constant, was clearly diminished relative to the control treated with the same volume of DMSO only (Figure 3). These observations seem to indicate fewer cells entering the S-phase, but S-phase cells still mostly progressing to G_2_/M phase, leading to the conclusion that DNA synthesis was not principally affected; **2** appeared to have a slightly more noticeable effect than **1** on the trypanosome cell cycle stages with the accumulation of about 40% DNA at the G_2_/M cell cycle phase after 10 h of incubation (Figure 3A) compared with about 35% observed in **1** (Figure 3B). The group treated with solvent only (DMSO) displayed the same configurations as an untreated control group assessed in parallel (not shown).

Microscopic examination of cell populations exposed for 12 h to 2× the respective EC_50_ concentrations of **1** or **2** prior to DAPI staining revealed swollen, misshapen cells, apparently containing replicated nuclear and kinetoplast DNA apparently stuck at G_2_/M phase as few cells contained more than one nucleus or kinetoplast (Figure 4), further confirming the above observed effects on DNA content analysis.

The number of nuclei (N) and kinetoplasts (K) per cell was monitored in the population by counting 500 cells and scoring them for the number of each DNA-containing organelle. Cells containing 2 nuclei and 2 kinetoplasts were further divided into early and late categories (2N2K-e, 2N2K-l) depending whether a clear ingression furrow for the longitudinal cell division could be observed. After 8 h of incubation with 1× EC_50_ of **1** or **2** cells started to appear that could not be classified in the regular categories and these were scored as ‘other’ (Figure 5). Observations were virtually identical for **1** and **2** throughout. Over the 24 h of the experiment, the number of 1N1K cells progressively and significantly increased relative to the control culture. Concurrently, there was a progressive decline in the number of 2N2K (early and late) cells, and the emergence of cells with an irregular DNA configuration as the cell structure deteriorated.

The morphology of the cells was studied in more detail by higher-resolution fluorescence microscopy with a DeltaVision Core microscope (Applied Precision, GE Healthcare Life Siences, Amersham, Bucks, UK). Figure 6 shows Delta Core images of cells stained with DAPI and MitoTracker after incubation for 8 h with 1× EC_50_ of **1** and **2**, and control unexposed cells. Some of the cells incubated for 8 h with either compound appeared to have larger, elongated nuclei compared to control cells (average nuclear volume increased to 152% (*p* = 0.0012) and 120% (*p* = 0.12) of control at 8 h of culture), but otherwise the cells had not noticeably changed as a result of this mild exposure, and staining with mitotracker did not reveal any mitochondrial damage at this point. The mitochondrion also appeared to be intact, an indication that 3-aminosteroids may not directly affect mitochondrial integrity. The untreated cells retained the typical trypomastigote shape throughout the duration of the experiment. After 8 h, some more highly damaged and even lysed cells began to appear (not shown).

### 2.6. Influence of 3-Aminosteroids on Mitochondrial Membrane Potential (MMP)

We next investigated the influence of **1** and **2** on MMP. The MMP of untreated cells remained fairly unchanged while the positive controls troglitazone and valinomycin caused a strong and rapid hyperpolarization and depolarization of MMP, respectively. The 3-aminosteroids were found to cause a slower but significant depolarization of the MMP (Figure 7A). Both compounds induced a gradual MMP depolarization effect, observed as early as 2 h of incubation, which was statistically significant for **2** (*p* < 0.05). From 4 h the decrease was significant for both treatments, and it continued to drop to approximately 20% at 12 h. At the chosen concentration of 2× EC_50_, **2** had a stronger MMP depolarizing effect than **1**, although this difference only showed minor significance at the 2 h point (*p* < 0.05), and may thus reflect a more rapid internalization of **2**.

### 2.7. Influence of 3-Aminosteroids on Intracellular ATP Levels

Further studies to assess any cause-effect relationship between *T. b. brucei* intracellular ATP level and the earlier observed effects were undertaken. The results showed that 3-aminosteroids rapidly and significantly (*p* < 0.05) decreased the ATP level in *T. b. brucei* (Figure 7B). The pattern closely followed that observed for the decrease in MMP, with **2** inducing a slightly more rapid and deeper decline than **1** for the concentrations chosen, but both otherwise following very much the same trend.

## 3. Discussion

The 3-aminosteroids showed impressive activities against the more virulent of the two haematic parasites, *Tc*-IL3000, than against *T. b. brucei*. Both parasites, in addition to *T. vivax*, cause Nagana in cattle but they can also cause serious losses in pigs, camels, goats, dogs and sheep. With **2** (IC_50_ = 1.75 µM) and **5** (IC_50_ = 1.62 µM) being the most active compounds against *T. b. brucei* under our experimental conditions, it can be inferred that 3-aminosteroids are moderately active against *T. b. brucei* while much higher levels of activity had previously been determined against the human pathogen, *T. b. rhodesiense*, using a different protocol [14]. It is important to note that, using an assay such as the resazurin-based one, EC_50_ values are not constants, but very much dependent on the duration of incubation and the cell density, and should be taken as a ranking that is internally consistent but not absolute. The protocol that is used as standard in the De Koning lab is particularly rigorous and tends to give substantially higher EC_50_ values for some classes of compounds than similar assays that use much lower cell densities.

Our observation regarding the structure-antitrypanosomal (SAR) of the 3-aminosteroids is that methylation of the 3-amino substituent and the stereochemistry at the C-3 position affects the activity against *T. b. brucei*, and particularly against *T. congolense*. For instance, the monomethylated 3-amino derivatives were more active than the non-methylated and dimethylated derivatives, as evidenced by comparing **2**, **7** and **9** with **1**, **8** and **10** respectively. The α-congeners were similarly more potent than the corresponding β-congeners, comparing **3** and **5** with **1** and **4** respectively, which might in part be attributable to their apparent uptake through the TbAT1/P2 transporter, as judged by the significantly reduced sensitivity to strains lacking this carrier. There was a very clear contribution of the additional pyrroline ring, formed from the C-18-*N*-C-20 bridge, to the activities of 3-aminosteroids against *Tc*-IL3000, as observed by comparing **1**–**5** vs. **6**–**10**. Interestingly, **10** (IC_50_ (*Tc*) = 0.045 µM), was ~2 times more active than the widely used veterinary drug diminazene aceturate [1] and 10 times more active than another well-known trypanocide, pentamidine, which is the standard treatment for early-stage HAT [32]. Hence, there will be a need to further explore its activity against *T. congolense* and carry out further SAR. The relatively higher activities and SI against *Tc*-IL3000 (up to 2130) of the pentacyclic compounds **6**–**10** strongly indicate a different mechanism for this aminosteroid series (whereas the very similar EC_50_ values for series **1**–**5** point for a similar mode of action against *T. congolense* and *T. brucei*), likely to be rooted in the biochemical and molecular differences between these species [33]. As a *T. congolense*-specific drug would not act against human sleeping sickness, and would have limited utility for AAT, where the infecting species is almost never known, we have here prioritised the general trypanocides **1** and **2**.

The emergence of resistant parasites in both the laboratory and the field suggest that these parasites are readily capable of adapting under drug pressure [34] although it has been found impossible to induce resistance to some natural compounds, including curcumin [35]. Resistance to the current trypanocidal drugs is a major bottleneck to the control of these severe vector-borne diseases [36,37] and trypanosomes resistant to almost all the available drugs have been identified [9,38]. In order to avoid introducing new trypanocides that have resistance profiles similar to that of the standard treatments, lead compounds should be subjected to cross-resistance studies. In this study, we found almost no significant differences in the susceptibility of various drug-resistant strains and the wild-type *T. b. brucei* to most of the compounds, and almost all observed resistance factors were below 2.5-fold. Significant increases in IC_50_ values were observed following treatment with compound **3** against TbAT1-KO (5.6), AQP2/3-KO (2.7), ISMR (3.2) and B48 (5.1), **5** against B48 (2.1) and **7** against TbAT1 (2.4), ISMR (2.9) and B48 (2.4), which indicated a somewhat reduced potency against these resistant strains. However, these figures are insignificant when compared with the resistance factors of these resistant strains against the standard drugs. Furthermore, there were no similar comparable tendencies for the other aminosteroids tested, emphasising that there is no cross-resistance problem for this pharmacological class as a whole, and that the 3β-aminosteroids, in particular, are unlikely to exhibit any cross resistance with existing first line HAT and AAT drugs such as pentamidine, melarsoprol, cymelarsan, isometamidium or diminazene, whose resistance mechanisms have been well characterised [22,24,37]. The data from the cross-resistance study indicate that 3-aminosteriods, being structurally dissimilar to any current trypanocides, are most unlikely to be internalized through the same drug transporters (possible exception for TbAT1/P2 and 3α-aminosteroids), or rely on the same intracellular target, and the presented data are consistent with this notion. Aside from the conclusion about resistance-associated transporters such as TbAT1/P2 [22,37], TbAQP2/HAPT1 [37,39] and the low-affinity pentamidine transporter LAPT [23,40], the absence of resistance with R0.8 indicates that the 3-aminosteroids do not act on the trypanosome’s cAMP signalling system [25], and the similar IC_50_ values against ISMR show that the test compounds have probably no direct effect on the F_o_F_1_ ATPase [24].

In the presence of continuous or time-limited (2 h) exposure to the 3-aminosteroids at concentrations ≥IC_50_, the growth pattern of *Tbb* 427WT was disrupted with no possibility of recovery, although complete clearance was not observed until 2–3 days after exposure. The early growth arrest in both cases and the complete clearance observed after 72 h of incubation suggest that 3-aminosteroids are trypanocidal and, like diamidines, prone to act very slowly but irreversibly [41]. The early effects on trypanosomes being irreversible once initiated is an undoubted advantage for in vivo efficacy as a short exposure time reduces potential cytotoxicity associated with the maintenance of peak drug plasma levels over an extended period of time. Of crucial importance for the treatment of AAT, moreover, is that it is in most cases impractical to have to administer multiple doses, over an extended time, to the infected animal. To introduce such a drug would be to practically ensure non-compliance, which would lead to relapses and promote rapid-onset resistance.

PI is a red fluorophore that is not taken up by live, intact trypanosomes but does enter, and bind to trypanosome nucleic acids, when the plasma membrane is compromised as a result of loss of cellular integrity [42,43]. The effects of 3-aminosteroids differ significantly from digitonin, a known cell membrane lysis agent, which causes rapid trypanosome cell permeabilization reaching maximum fluorescence in the first few minutes [44]. The much slower, concentration-dependent membrane permeabilization effect of the 3-aminosteroids suggests that these compounds do not directly cause membrane leakage by a physicochemical effect as their main mode of action, but rather that over time an increasing number of cells becomes sufficiently permeable to PI to elicit a fluorescence response. It should be noted that, while loss of cellular integrity obviously allows binding of PI to trypanosomal nucleic acids, not all cellular entry of PI is necessarily indicative of cell death having already occurred (as opposed to a small increase in membrane permeability), and that it does not reveal whether cell death occurs by necrosis or apoptosis, as some features of necrosis and apoptosis may be shared [45].

The slow trypanocidal effects of 3-aminosteroids as observed from the growth patterns and PI-based assay leaves the likely target of the compounds open. Thus, it was explored whether these compounds target aspects of the cell cycle, bearing in mind the unusual cell cycles of kinetoplastids, in which the nucleus, kinetoplast and basal body replicate and segregate in a coordinated fashion into daughter cells during cell division [46,47,48]. The PI-associated fluorescence after binding to *T. brucei* DNA is proportional to the amount of DNA per cell when analysed by flow cytometry [43,49]. This revealed a possible stall during the G_2_/M cell cycle phase, as the cells with double the DNA content of non-dividing cells progressively increased after treatment with either **1** or **2** (Figure 4 and Appendix A), although there was no increase, but rather a gradual decrease in the number of nuclei (Figure 6 and Figure 7). These findings suggest that DNA replication was unaffected, but that instead mitosis was inhibited, leading to significantly enlarged nuclei, with increased DNA content. In bloodstream form (BSF) *T. brucei*, inhibition of mitosis (M) prevents cytokinesis, although not re-replication of nuclear DNA (at G_2_ phase) and kinetoplast DNA (at S phase), or segregation of replicated kinetoplasts (at G_2_ phase), resulting in cells with a single enlarged nucleus and multiple kinetoplasts [50]. However, no cells with multiple kinetoplasts but only 1 nucleus were observed, other than the regular 1N2K cells that necessarily precede mitosis into 2N2K cells, and the proportion of those cells progressively and dose-dependently declined. Thus, it appears that it is the failure to divide the mitochondrially-located kinetoplasts that halts the cell cycle at that point and leads to the observed growth arrest and ultimate cell death.

This is not entirely unexpected as the 3-aminosteroids are likely to accumulate in the trypanosomal mitochondrion, by virtue of their lipophilicity and the cationic nature of their salts, formed in aqueous media, which provides mitochondrial targeting similar to previous lipophilic cations that have shown promise against African trypanosomes [27,51,52]. Mitochondrial destruction as their mode of action could not be validated in the present studies as mitochondrial integrity remained intact, judging by fluorescence microscopy following MitoTracker staining. However, the mitochondrial accumulation has been associated with many different trypanocidal activities of various compounds disrupting various mitochondrial functions. Examples are inhibitors of TAO coupled to lipo-cations [27]; di-cationic diamidines such as pentamidine and DB75 targeting multiple mitochondrial functions including kinetoplast DNA through accumulation to high concentrations [53,54,55]; choline-based di-cations disrupting mitochondrial functions [43]; bisphosphonium compounds inhibiting the activity of the *T. brucei* F_1_-ATPase [43]; disruption of kDNA replication by ethidium [56]; selective cleavage of kinetoplast DNA minicircles by diminazene and isometamidium [57]. Many of these drugs have been documented to bind to, or otherwise affect, the kinetoplast, but in general terms the principal consequence of the mitochondrial targeting of trypanocides is that they accumulate against a concentration gradient to a very high intra-mitochondrial concentration, driven by the mitochondrial membrane potential (pentamidine accumulates to mM concentrations [58]). This makes it likely that the drug can interfere with (multiple) biochemical processes in that space.

Although no direct mitochondrial damage was established after 8 h of incubation with 1× EC_50_ of **1** or **2**, this does not rule out either mitochondrial accumulation or a mitochondrial target for 3-aminosteroids. It has been found that kDNA replication requires mitochondrial proteins [59] that are encoded in the nuclear genome and imported into the mitochondrion. This import requires ATP and is proton-motive force-driven and hence depends on the mitochondrial membrane potential (MMP) [60]. The BSFs of *T. b. brucei* have a less elaborate mitochondrial metabolism than almost any other aerobic eukaryotic cell type; they lack the Krebs cycle enzymes as lack a cytochrome-dependent electron transport chain [52]. Therefore, they rely entirely on glycolysis instead of oxidative phosphorylation to generate ATP [61] and use the F_o_F_1_ ATPase to maintain mitochondrial membrane potential by pumping protons over the inner mitochondrial membrane (IMM) from the matrix to the intermembrane space [62]. We have previously reported that lipophilic bisphosphonium cations accumulate in the mitochondrion, cause a depolarisation of the IMM and reduce the cellular ATP content by inhibiting the F_1_ ATPase [52]. Although the lack of cross-resistance with the *T. brucei* ISMR strain, which derives its isometamidium resistance from a mutation in the γ-subunit of F_1_ [24], suggests that this ATPase is not the principal target of the aminosteroids, we found that, like the bisphosphonium salts and isometamidium, **1** and **2** rapidly depolarise the IMM, with significant decline in MMP in 4 h and 2 h, respectively. We likewise observed a rapid decline in cellular ATP levels, which could cause the IMM depolarisation, as this is directly maintained by ATP. Conversely, a functional mitochondrion is essential for ATP production in BSF trypanosomes, although the glycolysis and ATP production actually takes place in a unique trypanosomatid-specific organelle called the glycosome [63]. However, in order to maintain redox balance, the glycolysis intermediate glycerol-3-phosphate must be oxidized back to dihydroxyacetone phosphate by glycerol-3-phosphate dehydrogenase (G-3-PDH) in the mitochondrion, transferring 4 electrons to ubiquinol, which is subsequently oxidized to ubiquinone by TAO which then converts molecular oxygen to water [64,65]. This so-called glycerol-3-phosphate shuttle, and thus mitochondrial respiration, is dependent on the MMP [66], and without it BSF trypanosomes generate just one ATP per glucose, explaining the rapid and extreme decline in cellular ATP levels also observed with the F_o_ inhibitor oligomycin [67]. Thus, MMP and ATP content are linked, as also observed for the bisphosphonium compounds [52], but whereas those compounds caused an increase in G_1_-phase cells, the 3-aminosteroids caused an increase of cells arrested in G_2_/M phase.

Overall, we conclude that 3-aminosteroids act on the trypanosomal mitochondrion, by which they cause a strong reduction of the MMP, and ATP levels, and prevent the division of kinetoplast, resulting in irreversible G_2_/M phase cell cycle arrest and eventual cell death. Further studies should assess ultrastructural changes in the kinetoplast and the integrity of kDNA.

## 4. Materials and Methods

### 4.1. Parasites, Cell Lines and Cultures

The parasites used for this study include *Trypanosoma congolense Tc*-IL3000 (Savannah-type strain IL 3000) [27], *T. brucei brucei* (BSF trypomastigotes *T. b. brucei* strain 427 Lister WT (*Tbb* 427WT), MiTat 1.2/BS221) and six resistant cell lines of *T. brucei brucei* BSF trypomastigotes which have been described [22,23,24,25,28,68]. All the *T. b. brucei* strains were cultured in standard Hirumi’s modified Iscove’s medium 9 (HMI-9), supplemented with 10% heat-inactivated foetal bovine serum (FBS), 14 µL/L β-mercaptoethanol, and 3.0 g sodium hydrogen carbonate per litre of medium adjusted to pH 7.4, *Tc*-IL3000 strains were cultured in Dulbecco’s minimum essential medium (MEM) with β-mercaptoethanol, sodium pyruvate, sodium bicarbonate and supplemented with fresh goat serum. All the parasites were cultured in vented flasks at 37 °C in a 5% CO_2_ atmosphere and were passaged every 72 h.

### 4.2. Determination of In Vitro Anti-Trypanosomal Activity of Compounds

All tested compounds were isolated from *Holarrhena africana* A. DC. (a synonym of *H. floribunda* (G. Don) T. Durand & Schinz) (Apocynaceae) and identified as described in our previous communication [14]; ^1^H- and ^13^C-nuclear magnetic resonance (NMR) spectra to assess the identity and purity can be found in the supplementary information of [14]. The in vitro activities of **1**–**10** against BSF trypanosomes were determined using a resazurin-based assay protocol as described [24,27,42] using cell densities adjusted to 2 × 10^5^ cells/mL for *T. b. brucei* and 5 × 10^5^ cells/mL for *Tc*-IL3000. Cells were exposed to at least 11 doubling dilutions of test compound starting at 100 µL, with the last well in the row receiving medium without test compound as a drug-free control. The trypanosome culture and test compounds were incubated for a period of 48 h at 37 °C for *T. b. brucei* and 32 °C for *Tc*-IL3000 under a 5% CO_2_ atmosphere. In all the assays, pentamidine and diminazene aceturate were used as positive controls. Fluorescence was measured in the 96-well plates with a FLUOstar Optima (BMG Labtech, Aylesbury, Bucks, UK) at wavelengths of 544 nm for excitation and 590 nm for emission. IC_50_ values were calculated by non-linear regression using an equation for a sigmoidal dose-response curve with variable slope (Prism 5.0, GraphPad Software, San Diego, CA, USA).

### 4.3. Effect of Treatments on Growth Patterns of T. b. brucei after Short- and Long-Term Exposure

Varying concentrations of **1** and **2** were tested on trypanosomes in order to determine in vitro cell growth patterns and time-to-kill. Trypanosomes at their mid logarithmic phase of growth were taken from cultures and cell density was determined using a haemocytometer (Camlab, Over, UK). The cell density was adjusted to 2 × 10^5^ cells with fresh HMI-9 medium and the fresh cultures were incubated with varying concentrations of **1** and **2** at 37 °C under a 5% CO_2_ atmosphere. For the limited time of exposure assay, the cells were similarly incubated, but after 2 h of exposure to the test compounds, the cells were washed twice with fresh HMI-9 media, re-seeded at the same cell density and again incubated under the same conditions. At predetermined intervals, cell counts were taken from each culture, typically for up to 72 h, and the counts were used for plotting the growth curves.

### 4.4. Determination of Speed of Action by Propidium Iodide (PI) Assay

100 μL of 18 µM PI in HMI-9 was added to each well of a 96-well plate. In the first column of the wells, 200 μL solutions of test compound, also in 18 µM PI/HMI-9, was added and serially diluted across the wells. Wells receiving only media without test compound served as drug-free controls. To each well was added 100 μL of 2 × 10^6^
*T. brucei* 427WT cells/mL in HMI-9. Wells containing the same final concentration of 9 μM PI in HMI-9 but no cells served to record background fluorescence. The plates were incubated in a FLUOstar OPTIMA fluorimeter at 37 °C with 5% CO_2_ atmosphere, and the fluorescence was recorded at 544 nm excitation and 620 nm emission for 120 cycles at 180 s per cycle.

### 4.5. Determination of Intracellular ATP Level

Intracellular ATP was measured using a Molecular Probes ATP Determination Kit (Molecular Probes Inc., Cambridge, UK) following the manufacturer’s protocol. Trypanosome culture density was adjusted to 10^7^ cells/mL and incubated with **1** or **2**. At predetermined time intervals, 1 mL parasites culture was taken and centrifuged for 10 min at 2500× *g*, 4 °C. The resulting pellets were lysed by sonication and cell debris was removed using a refrigerated microcentrifuge (SciQuip, Newtown, UK) at 12,000× *g* for 10 min at 4 °C. The resultant supernatant was frozen in liquid nitrogen and stored at −80 °C. For the analysis, 90 μL reaction solution from the master mix was added to each designated well of a 96-well plate and the background luminescence was recorded using a FLUOstar OPTIMA fluorimeter; then 10 μL of each sample was added to the well and incubated for 15 min at 28 °C and the luminescence was recorded. A standard curve was made using serial dilution of varying concentrations of ATP (0.5–500 pM) to calculate the ATP concentrations in the samples.

### 4.6. Effects of Treatments on Cell Morphology

A *T. b. brucei* culture was adjusted to 2 × 10^5^ cell/mL and incubated with **1** and **2** for the duration of the assay. The MitoTracker red CMXRos (New England Biolabs Ltd, Hitchin, UK) used for this assay was prepared from a stock solution of 100 μM in DMSO. From this solution, 1 μL was added to 1 mL of cell sample taken from the cell culture at each time point to make a final concentration of 100 nM of MitoTracker. The sample was then incubated at 37 °C and 5% CO_2_ for 5 min. The incubated sample was washed in filter-sterilized 1× phosphate buffered saline (PBS) and spun at 2600× *g* for 10 min (4 °C). After the final wash step, the sample was re-suspended in 1 mL of 1× PBS. A 50 μL of the re-suspended cells were spread onto a glass microscope slide (CamLab), previously coated with 50 μL of poly-L-lysine (Sigma-Aldrich, Gillingham, Dorset, UK); and were left to air dry, the cells were then fixed in 4% paraformaldehyde/1× PBS for 10 min at room temperature. The slides were covered with 1 mL of PBS and allowed to rehydrate for 10 min after which it was then allowed to evaporate but not completely. A drop of Vectashield anti-fade mounting medium with DAPI (Vector Laboratories, Burlingame, CA, USA) was added to the slides and spread by a coverslip (CamLab, Over, UK); the coverslip was then applied and the edges were sealed with nail varnish. Slides were observed for cell morphology under an Axioskop microscope (Image Solutions. Preston, UK) using softWoRx software (Suite 2.0, Applied Precision, GE Healthcare, Amersham, Bucks, UK), while DNA configuration was assessed using a Zeiss Axioplan microscope (Cambridge, UK) using Hamamatso digital camera and Openlab software. For detailed observation of single DAPI-stained cells and the measurement of nuclear volumes, Z-stacks of cells labelled for DAPI and MitoTracker were acquired using an Olympus UPLSAPO 100× oil (1.40 NA) objective on a DeltaVision Core microscope (Applied Precision, GE Healthcare Life Siences, Amersham, Bucks, UK) attached to a CoolSNAP HQ2 CCD camera (Photometrics, Tucson, AZ, USA). Deconvolution was performed using SoftWoRx Suite 2.0), and later data was analysed using FIJI [69] and/or IMARIS (Bitplane, Belfast, UK).

### 4.7. Determination of Mitochondrial Membrane Potential

FACS was employed in the determination of the changes in MMP due to exposure of *T. b. brucei* to **1** and **2** by TMRE dye incorporation as previously described [43,52]. The cell density was adjusted to 10^6^ cells/mL and incubated with **1** or **2** at 32 °C under 5% CO_2_. At predetermined intervals, 1 mL of sample was washed and re-suspended in 1 mL PBS containing 200 nM of TMRE in FACS tubes, followed by incubation at 37 °C for 30 min, and analysed by a Becton Dickinson FACS Calibur (Franklin Lakes, NJ, USA) using a FL2-heigth detector and CellQuest (Becton Dickinson) and FlowJo software (V.10, Ashland, OR, USA). Valinomycin (100 nM; Sigma-Aldrich) and troglitazone (10 µM; Sigma-Aldrich) were employed as mitochondrial membrane depolarizer and hyperpolarizer control, respectively.

### 4.8. Determination of Cell Cycle and DNA Content Assay

FACS analysis was performed to study the effects of **1** and **2** on cell cycle of trypanosomes, essentially as described previously [49] with the following modifications. Approximately 2.5 × 10^6^ cells were harvested by centrifugation at 1500× *g* for 10 min at 4 °C and washed once in 1× PBS by centrifugation (3000× *g* for 5 min). The cells were re-suspended in 1 mL of 70% ethanol and 30% 1× PBS. The tubes with the cells were left at −20 °C to fix overnight in the dark and the samples were subsequently washed once with 1 mL 1× PBS, re-suspended in 500 µL 1× PBS containing 10 μg/mL PI and 100 μg/mL RNase A and incubated at 37 °C for 45 min before FACS analysis. The samples were analyzed by a Becton Dickinson FACSCalibur and PI incorporation was measured using the FL2-Area detector and CellQuest software. The data obtained were analyzed using FlowJo software (V.10).

## Figures and Tables

**Figure 1 molecules-24-00268-f001:**
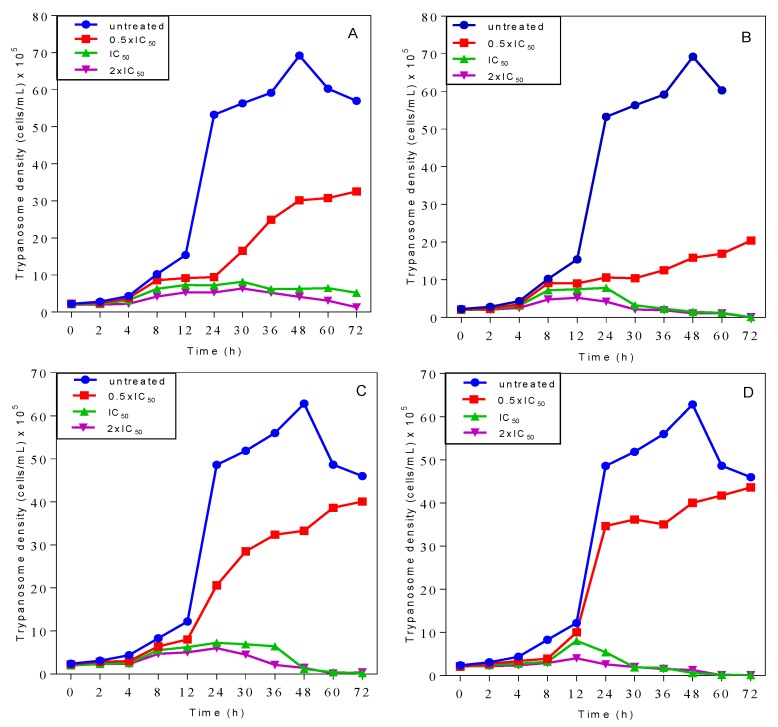
Growth curves of wild-type *T. b. brucei* grown in the continuous presence of **1** (**A**) or **2** (**B**) and in Hirumi’s modified Iscove’s medium (HMI-9) standard growth media after a short incubation period (2 h) with **1** (**C**) or **2** (**D**).

**Figure 2 molecules-24-00268-f002:**
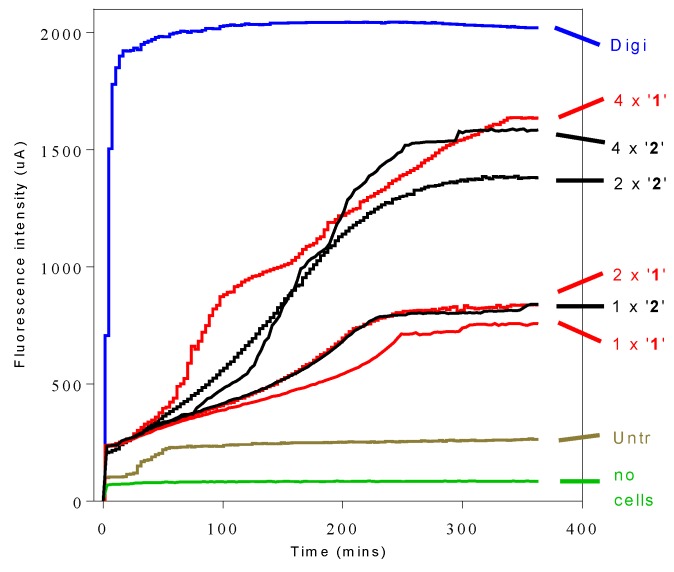
Speed of action profile of 3-aminosteroids. The profiles represent fluorescence intensity of nucleic acid–propidium iodide (PI) complex for *T. b. brucei* incubated with **1** (red traces) and **2** (black traces) at concentrations 1×, 2× and 4× IC_50_ values and controls: Digi, digitonin (blue trace; positive control for 100% cellular disruption); Untr, PI + untreated cells only (Tan trace); and no cell, PI background without cells (green trace, ‘no cells’).

**Figure 3 molecules-24-00268-f003:**
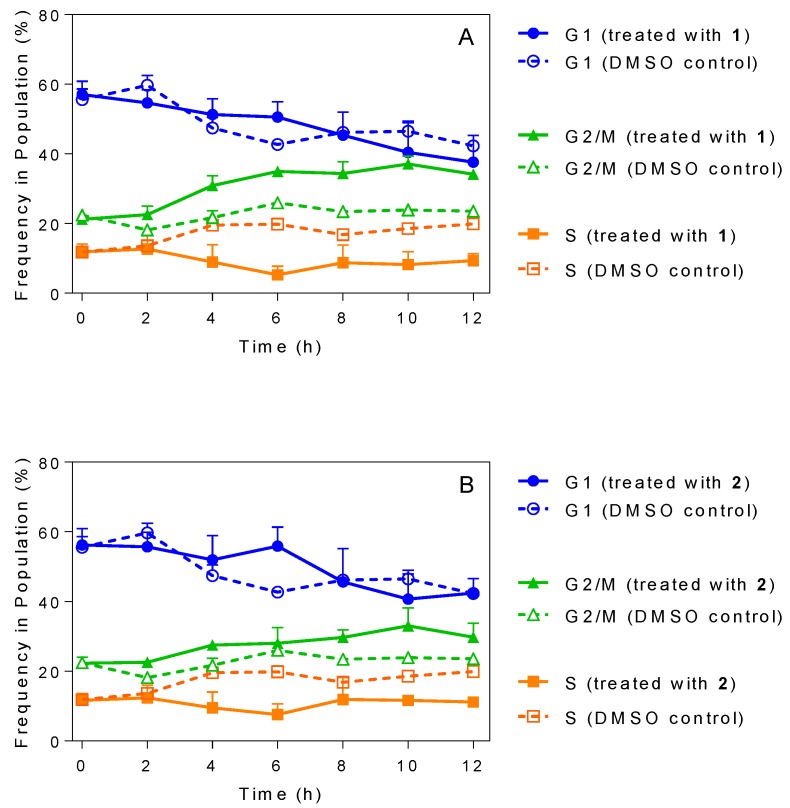
DNA content of *T. b. brucei* cells for (**A**) = cells treated with 4× EC_50_
**1**; (**B**) = cells treated with 4× EC_50_
**2**; error bars represent SD; data for frequency in population. Graphs depict DNA content collected by flow cytometry. The percentages of each cell cycle stage were plotted as a frequency of the total cell population (%) over 12 h.

**Figure 4 molecules-24-00268-f004:**
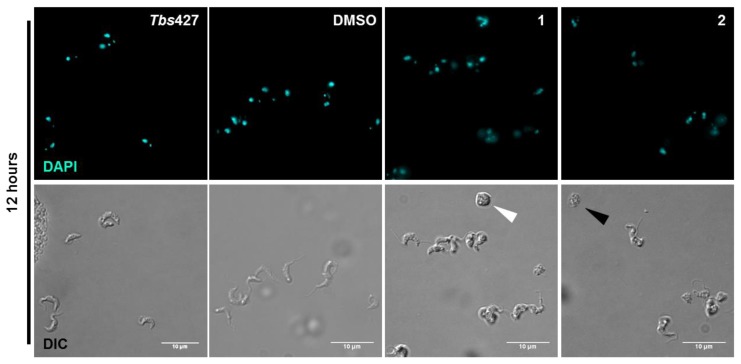
Morphological examination of DAPI-stained treated (with DMSO, **1** or **2**) and untreated *T. b. brucei* 427WT cells; the marked cells (black and white arrow heads) were representative cells with characteristic effects on size, morphology, motility and configuration post-treatment with **1** or **2**.

**Figure 5 molecules-24-00268-f005:**
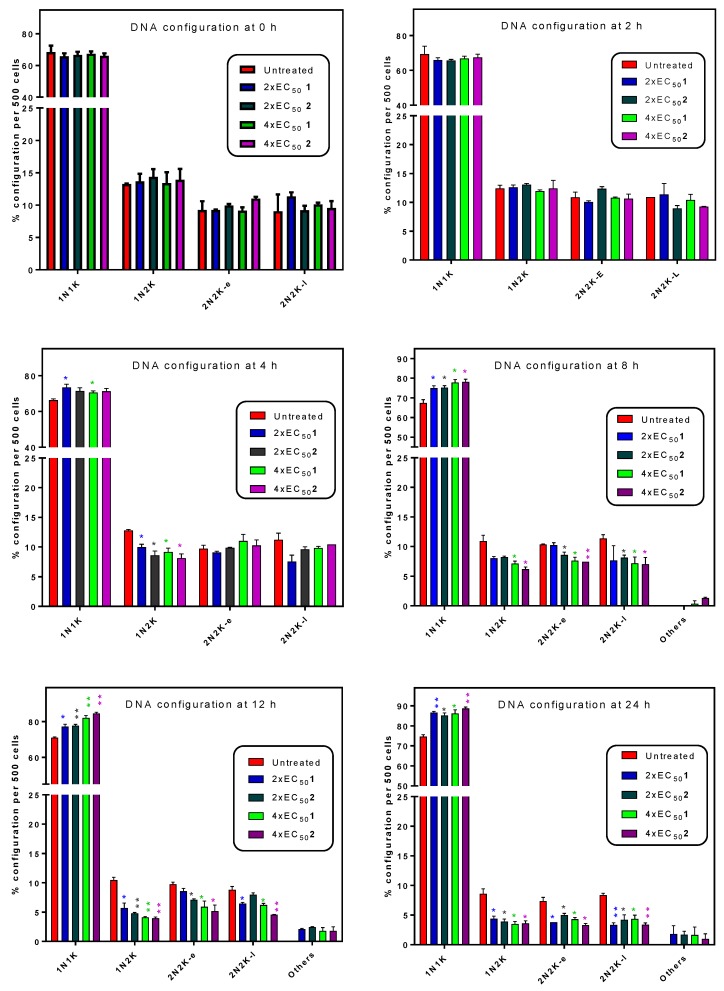
Cell configuration in DAPI stained cell populations. Samples were taken at the indicated times from cultures incubated with or without test compounds **1** and **2**, stained with the nucleic acid dye DAPI and observed by fluorescence microscopy. For each sample, 500 cells were scored in categories for the number of kinetoplasts (K) and nuclei (N), with the sub-classification early (e) indicating the absence of an ingression furrow for cell division, and late (l) indication a clear presence of an ingression furrow. Each bar represents the average and SD of two independent experiments. In each category, the treated samples were statistically compared to the treated sample by unpaired *t*-test. * *p* < 0.05; ** *p* < 0.01. For the category ‘others’ no statistical comparison could be performed as no ‘others’ were observed in the control culture. Observation and scoring of cells were done using a Zeiss Axioskop fluorescence microscope.

**Figure 6 molecules-24-00268-f006:**
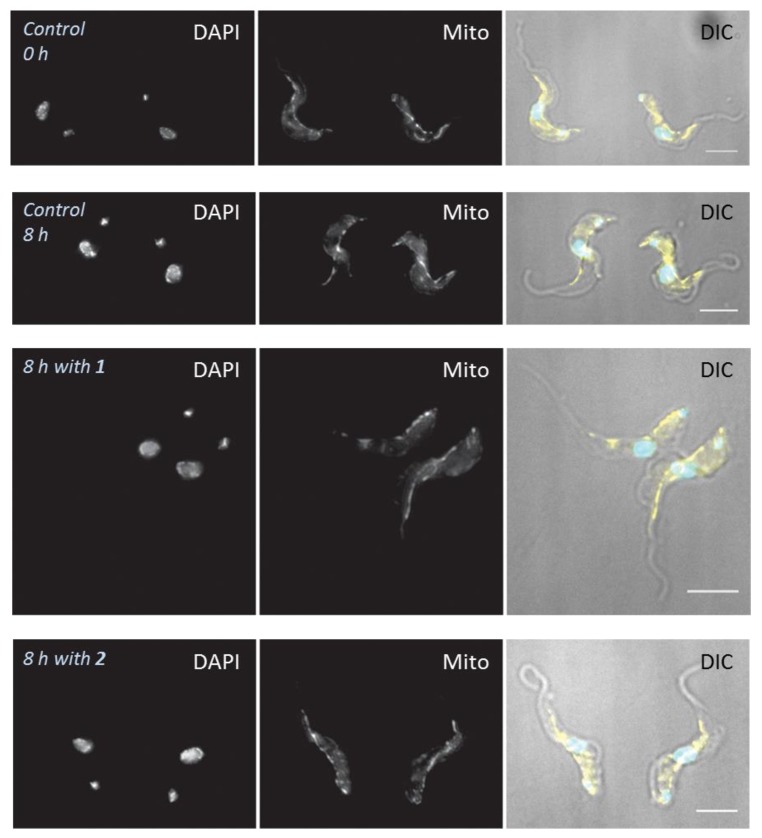
High-resolution fluorescence microscopy of individual cells observed with wide-field deconvolution fluorescence microscope. In the overlay, cyan is DAPI, yellow is MitoTracker. Representative images, taken of random cells (blind study). Scale bars are 5 µm.

**Figure 7 molecules-24-00268-f007:**
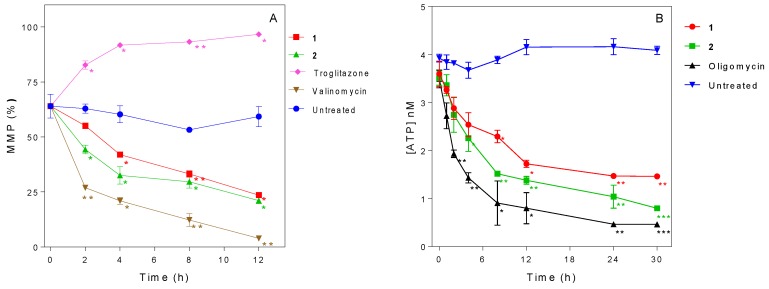
Effects of 3-aminosteroids **1** and **2** on mitochondrial membrane potential (**A**) and intracellular ATP level (**B**) of *Tbb* 427WT, each at 2× EC_50_ concentration. * *p* < 0.05; ** *p* < 0.01; *** *p* < 0.001 indicate significant difference from the corresponding untreated control at the same time point (unpaired Student’s *t*-test (n = 3)); error bars = ± SD.

**Table 1 molecules-24-00268-t001:** In vitro activities of some 3-aminosteroids against wild-type *T. b. brucei* (*Tbb* 427WT) and *Tc*-IL3000.

Compounds		*Tbb* 427WT		*Tc*-IL3000	
		IC_50_, µM	S. I.	IC_50_, µM	S. I.
**1**	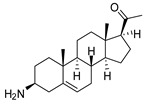	4.53 ± 0.16	1.1	20.8 ± 0.84	0.3
**2**	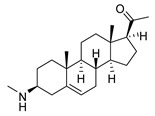	1.75 ± 0.17	1.4	3.78 ± 1.96	0.7
**3**	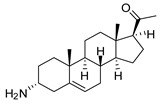	3.41 ± 0.17	4.6	9.49 ±0.30	1.7
**4**	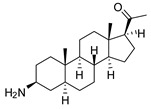	4.00 ± 0.24	4.3	2.93 ± 0.33	5.9
**5**	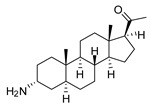	1.62 ± 0.09	10.5	6.08 ± 4.97	2.8
**6**	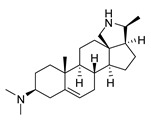	>100	<0.5	1.07 ± 0.32	47.2
**7**	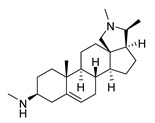	3.42 ± 0.21	8.0	0.22 ± 0.35	123.5
**8**	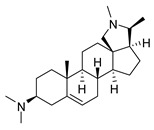	3.51 ± 0.10	17.4	1.65 ± 0.92	37.2
**9**	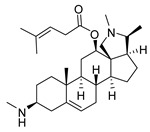	2.20 ± 0.08	6.5	0.22 ± 0.13	65.9
**10**	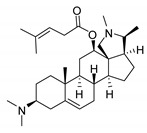	4.79 ± 0.81	20.4	0.045 ± 0.03	2130
	Pentamidine	0.004 ± 0.001	-	0.448 ± 0.28	-
	Diminazene	0.066 ± 0.002	-	0.093 ± 0.01	-

Data are average of three independent determinations and expressed as mean IC_50_ ± standard deviation (SD); selectivity indices (S.I.) = cytotoxic (IC_50_) on L6 cell/IC_50_ target cell; the IC_50_ values for cytotoxicity against L6 rat skeletal myoblasts as well as against *T. b. rhodesiense* have been reported elsewhere [14].

**Table 2 molecules-24-00268-t002:** In vitro activities and cross resistance profile of selected 3-aminosteroids against some *Tbb* 427WT-derived resistant cell lines.

Compound	*Tb*AT1-KO		AQP1-3-KO		AQP2/3-KO		ISMR		B48		R0.8	
	IC_50_ (µM)	RF	IC_50_ (µM)	RF	IC_50_ (µM)	RF	IC_50_ (µM)	RF	IC_50_ (µM)	RF	IC_50_ (µM)	RF
**1**	3.2 ± 0.6	0.7	3.6 ± 0.2	0.8	6.6 ± 0.5	1.4	7.2 ± 0.8	1.6	4.4 ± 0.2	0.9	2.6 ± 0.3	0.6
**2**	1.6 ± 0.2	0.9	1.5 ± 0.2	0.9	2.7 ± 0.2	1.5	2.9 ± 0.9	1.7	2.3 ± 0.4	1.3	1.1 ± 0.1	0.6
**3**	19 ± 0.3 ***	5.6	4.7 ± 0.2	1.4	9.1 ± 1.0 **	2.7	11.0 ± 0.5 ***	3.2	17.4 ± 0.8 ***	5.1	5.3 ± 0.6	1.6
**4**	5.5 ± 0.4	1.4	2.7 ± 0.3	0.7	5.8 ± 0.4	1.4	8.0 ± 0.5 *	2.0	7.8 ± 0.7	1.9	5.9 ± 0.6	1.5
**5**	1.7 ± 0.1	1.0	1.4 ± 0.1	0.9	2.6 ± 0.4	1.6	2.9 ± 0.3	1.8	3.5 ± 0.05 ***	2.1	1.7 ± 0.6	1.1
**6**	>100	-	>100	-	>100	-	>100	-	>100	-	>100	-
**7**	8.2 ± 0.2 ***	2.4	1.7 ± 0.2	0.5	3.4 ± 0.2	1.0	10 ± 0.4 ***	2.9	8.2 ± 0.1 ***	2.4	6.0 ± 0.9*	1.8
**8**	4.2 ± 0.1	1.2	4.6 ± 0.2	1.3	4.1 ± 0.1	1.2	2.7 ± 0.1	0.8	3.5 ± 0.4	1.0	3.4 ± 0.2	1.0
**9**	2.1 ± 0.1	0.9	1.8 ± 0.2	0.8	3.2 ± 0.3	1.4	3.7 ± 1.0	1.7	3.4 ± 0.2	1.6	2.6 ± 0.2	1.2
**10**	3.3 ± 0.2	0.7	1.4 ± 0.2	0.3	3.3 ± 0.1	0.7	6.2 ± 0.8	1.3	4.6 ± 0.1	0.9	2.6 ± 0.1	0.5
PMD ^a^	0.0077 ± 0.0002 **	2.0	0.058 ± 0.003 ***	15	0.046 ± 0.002 ***	12	0.0065 ± 0.0010 *	1.7	0.55 ± 0.07 **	143	0.0072 ± 0.0004 **	1.9
DA ^b^	0.47 ± 0.02 ***	7.0	ND	-	ND	-	1.1 ± 0.1 ***	17	0.74 ± 0.04 ***	11	ND	-

^a^ PMD, pentamidine; ^b^ DA, diminazene aceturate; RF = resistance factor = IC_50_ (resistant strain)/IC_50_ (*Tbb* 427WT). RF ≈ 1 or < 1 indicates no prospect for cross resistance; RF >> 1 indicates the possibility of cross resistance; RF > 1 indicates reduced potency RF < 1 indicates increased potency. *p*-values were calculated using the unpaired Student’s *t*-test (n = 3); * *p* < 0.05; ** *p* < 0.01; *** *p* <0.001 are significantly different from wild-type control values in Table 1. ND, not determined.

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
