# Peer review of "Potent Antitrypanosomal Activities of 3-Aminosteroids against African Trypanosomes: Investigation of Cellular Effects and of Cross-Resistance with Existing Drugs"

_molecules, 2019, doi:10.3390/molecules24020268_

Round 1

Reviewer 1 Report

Animal African trypanosomiasis is a serious problem in many countries of Africa. Thomas J. Schmidt and co-workers focused on in-depth anti-AAT examination of 10 natural 3β-aminosteroids. Despite only one compound (10, IC50 = 0.045 ± 0.03 µM) showed significant activity this is an important body of new results and data significantly contributing to the general field of antitipanosomal activity compounds. Manuscript is well written and , as to me, the interesting finding is that investigated compounds affect mitochondria of parasites. 

Author Response

We thank this reviewer for the very positive assessment.

Reviewer 2 Report

The presented article is devoted to finding new substances for the treatment of Animal African trypanosomiasis. This is very important because Animal African trypanosomiasis causes great damage to livestock. Currently used drugs for the treatment of Animal African trypanosomiasis are becoming less and less effective. The authors conducted research on the study of the trypanocidal activities, and the mode of action of selected 3-aminosteroids against Trypanosoma brucei brucei and T. congolense. Work performed at a high level and causes a favorable impression.
However, there are two important notes to the manuscript:
1. From the text of the work, it remains not entirely clear why the authors chose substances 1 and 2 to study the mechanism of action of 3-aminosteroids and did not take into account, for example, substance 9 or 10, which contain a pyrrole cycle.
2. The authors in the section «Materials and Methods» should provide the procedure for the synthesis of 3-aminosteroids with a demonstration of the physicochemical characteristics of the target substances and graphical NMR spectra in the Supporting Information.
 The manuscript can be accepted for publication after making corrections.

Author Response

Reviewer 2

The presented article is devoted to finding new substances for the treatment of Animal African trypanosomiasis. This is very important because Animal African trypanosomiasis causes great damage to livestock. Currently used drugs for the treatment of Animal African trypanosomiasis are becoming less and less effective. The authors conducted research on the study of the trypanocidal activities, and the mode of action of selected 3-aminosteroids against Trypanosoma brucei brucei and T. congolense. Work performed at a high level and causes a favorable impression.
However, there are two important notes to the manuscript:

1. From the text of the work, it remains not entirely clear why the authors chose substances 1 and 2 to study the mechanism of action of 3-aminosteroids and did not take into account, for example, substance 9 or 10, which contain a pyrrole cycle.

The focus of this current study was not so much on structure-activity relationships. Mechanistic experiments such as those presented here are relatively laborious and time consuming. We decided to concentrate on structurally simple compounds such as 1 and 2 in order to gain first insights into their mechanism of action. Two structurally similar compounds were chosen in order to find out whether they do act by the same mechanism and whether this mechanism is related to the overall activity observed. This is obviously the case as we could show in this work, since compound 2 is somewhat more active in most of the biochemical experiments as it is also in the whole cell assay. The data obtained provides an important first insight into the mechanism of action. Further studies on the more complex pentacyclic congeners will certainly have to follow in order to obtain more knowledge on structure-activity relationships but we would rather perform such studies after full elucidation of the mechanism of action and identification of the molecular target of these structurally more simple compounds.

2. The authors in the section «Materials and Methods» should provide the procedure for the synthesis of 3-aminosteroids with a demonstration of the physicochemical characteristics of the target substances and graphical NMR spectra in the Supporting Information.

The compounds are not synthetic but all natural products isolated from a plant. Their isolation and identification (along with all necessary analytical data and NMR spectra) has been described in detail in reference [14]. Since this is clearly mentioned, and properly referenced, already in the introduction of the manuscript, it is not necessary to repeat this in the present communication. However, a short statement on this fact has been added to the Materials and methods under section 4.2 to make it easier for the reader.

The manuscript can be accepted for publication after making corrections.

We cordially thank both reviewers for the positive assessment.